# Multi-Feature Unsupervised Domain Adaptation (M-FUDA) Applied to Cross Unaligned Domain-Specific Distributions in Device-Free Human Activity Classification

**DOI:** 10.3390/s25061876

**Published:** 2025-03-18

**Authors:** Muhammad Hassan, Tom Kelsey

**Affiliations:** School of Computer Science, University of St. Andrews, St. Andrews KY16 9SX, UK; mh353@st-andrews.ac.uk

**Keywords:** cross-domains, domain-generalization, multi-source unsupervised domain adaptation, combined-source unsupervised domain adaptation, channel state information, diverse domains, device-free sensing

## Abstract

Human–computer interaction (HCI) drives innovation by bridging humans and technology, with human activity recognition (HAR) playing a key role. Traditional HAR systems require user cooperation and infrastructure, raising privacy concerns. In recent years, Wi-Fi devices have leveraged channel state information (CSI) to decode human movements without additional infrastructure, preserving privacy. However, these systems struggle with unseen users, new environments, and scalability, thereby limiting real-world applications. Recent research has also demonstrated that the impact of surroundings causes dissimilar variations in the channel state information at different times of the day. In this paper, we propose an unsupervised multi-source domain adaptation technique that addresses these challenges. By aligning diverse data distributions with target domain variations (e.g., new users, environments, or atmospheric conditions), the method enhances system adaptability by leveraging public datasets with varying domain samples. Experiments on three public CSI datasets using a preprocessing module to convert CSI into image-like formats demonstrate significant improvements to baseline methods with an average micro-F1 score of 81% for cross-user, 76% for cross-user and cross-environment, and 73% for cross-atmospheric tasks. The approach proves effective for scalable, device-free sensing in realistic cross-domain HAR scenarios.

## 1. Introduction

The vision of a smart city relies on interactive data dissemination among smart devices through human interaction. Human involvement in commanding machines has traditionally involved manual switching. In recent years, we have entered a developed era of audiovisual techniques involving little or no human obtrusion, bringing the concept of an ultra-modern society closer to reality. Audiovisual and sensor-based networks can monitor an environment continuously for long periods with an unobtrusive sensing mechanism. This unobtrusive sensing forms a key contribution to our ability to recognize human gestures by using smart technologies, motivating the role of an advanced field of sensing, termed human activity recognition (HAR). HAR-based systems are helpful in propagating instructions from humans to computers through gestures/activities without making physical contact. Initial research on HAR-oriented systems mainly concerns audiovisual-based technologies using static or moving image data captured through cameras [1,2,3]. This kind of recognition task has serious privacy issues since the collected data are interpreted into human-perceivable form. In- and on-body sensors have also been intensively studied for human motion tracking in coarse- and fine-grain applications. However, they have technical requirements, in particular sensor-mounted devices such as accelerometers, and gyrosensors, and rely on a pre-installed setup with sensing devices attached to or placed inside a person’s body [4,5,6,7,8,9]. This is not a convenient way of monitoring, and constant examination is impossible because users struggle to carry such cumbersome devices all the time [10]. Wi-Fi-based systems have led to a breakthrough in the field of sensing due to their ubiquitous availability for data communication. This scenario has been successfully leveraged for a wide range of operations including, but not limited to, health monitoring [11,12,13,14], fall detection [15,16], human gesture and location identification [17,18,19,20], activity recognition [21,22,23], object detection [24,25], and humidity estimation [26]. Inspiration for this modern device-free sensing comes from (i) an easy and convenient way of sensing the environmental impacts of an activity with the help of a multi-path radio wave propagation, (ii) infrastructure-less operation with readily and extensively accessible radio devices, and (iii) high adherence to privacy and governance standards, as the wireless sensor data are complex and cannot be easily understood by the intended or unintended users.

Over the past few years, wireless-fidelity (Wi-Fi) sensor-based networks have seen significant demand and rapid growth for transmitting data via radio signals. This is due to their ease of large-scale deployment and high throughput capabilities. Recently, these sensor networks have been utilized for tracking human motion in laboratory environments using device-free wireless sensing (DFWS) technology. Wi-Fi communication is a multi-path propagation of radio waves. These radio signals arrive at the receiving antenna after following several paths with different time delays and undergo reflection, refraction, and scattering by the static and moving objects present in a sensing area. The radio signals emitted by these devices experience fluctuations caused by movement within the sensing environment. These fluctuations create distinct activity patterns at the receiving end which can be leveraged for motion detection.

However, the effectiveness of this technology depends on several factors, including sensitivity to changes in the sensing environment, the presence of previously unseen subjects, the ability to track multiple targets, data collection configurations, and the specific activities being monitored [27]. In other words, differences in domain distributions between the training and testing phases arise due to these variations, which are highly likely to occur in real-world scenarios. These are called the environmental effects of surroundings on radio waves [28]. Static objects produce persistent impacts that remain consistent unless the sensing environment changes, whereas the impact of moving objects creates a unique pattern associated with a particular activity. These systematic fluctuations establish a solid foundation for training a deep model to learn a distinctive mapping of an activity for its classification via channel state information (CSI). The CSI defines wireless channel propagation characteristics that include the consequences of fading and scattering on the system [29]. It is a complex representation of a unique pattern of static and dynamic vectors’ movement in a 3D space which can be analyzed by vector magnitude and phase in a complex polar form [30]. Since radio signals are heavily dependent upon environmental specifications, the performance of such trained deep models is relevant to these conditions. There are limitations and challenges which motivate novel active research in similar domains. The recorded CSI data are assumed to be generated for a particular environment with certain characteristics, such as the position of surrounding objects, which are mainly responsible for constructing static vectors. If their positions are changed from training to testing data, network performance suffers from degradation [31,32]. Interference caused by other moving objects apart from the target movement in the sensed area can lead to changes in the recorded CSI from training to the testing data because it may produce additional phase shifts or alter the magnitude of the dynamic vector and affect the system overall accuracy [30]. Hardware used in the transceiver antennas produces noises such as sampling frequency offset (SFO) which may differ by the variation of hardware from different vendors. Thus, hardware noises can modify the results and ultimately the accuracy of the system [33]. Interference generated by other Wi-Fi devices and microwave appliances nearby is another cause of desensitization of the recorded CSI data [34]. The CSI measurements are also sensitive to atmospheric conditions and can show dissimilarities in recorded data for an activity collected at different times or days of the week [35]. Since multiple factors can restrict a system’s performance in a controlled environment, every time a system undergoes any change, it needs to be retrained with new data samples for mapping the CSI variations according to the recent modifications. Collecting new data samples is impractical, as it takes heavy data storage, maintenance, human labor, and time. Retraining a model from scratch is also computationally expensive. We therefore seek a concrete solution that can help to generalize this field for more practical applications.

In this paper, we attempt for the first time to apply multi-source domain adaptation to an unsupervised model using Wi-Fi-based CSI data for cross-domain applications. Given the fact that domain shifting has the extra burden of generating, migrating, and annotating sufficient data, we take advantage of multiple scarce source domains to map their feature spaces to a target domain. Most of the public CSI datasets are collected in such a way that they cannot be used for cross-domain applications due to limited data samples per domain (particular user/environment or a specific day of the week). However, the proposed model overcomes the problem by successfully leveraging the data from diverse domains and aligning the distributions related to each specific source domain with a target domain. In addition, we train domain-specific classifiers by using domain-invariant representations acquired from diverse source domains. When these trained classifiers are used to predict target classes, they might classify them differently due to distribution mismatch near the decision boundaries. Ultimately, we align the decision boundaries of each classifier for the target domain. Figure 1 illustrates the workflow of our proposed methodology, detailing the process from collecting raw CSI measurements to classifying the activities being performed. (1) We first extract the raw CSI magnitude values. (2) We normalize them to a scale of [0–255] RGB scale. (3) We apply PCA for selecting features with maximum variation. (4) We use t-SNE embedding to find local relationships within high-dimensional data by projecting them into a lower-dimensional space. (5) Spectrograms are generated leveraging STFT and are fed to the model as input. (6) The model use such spectrograms generated from diverse sources and transforms the model to adopt a new target domain. (7) Finally, after being optimized, the model classifies activities on this new domain.

The motivation behind our proposed model stems from the advantages of leveraging diverse source distributions. Models trained with multiple sources develop richer feature representations and exhibit greater robustness when adapting to unseen target domains. This is because having multiple sources increases the likelihood of overlapping features with the target domain, facilitating smoother adaptation and reducing overfitting to a single source. Moreover, extracting common domain-invariant representations becomes more challenging in a SUDA (single-source unsupervised domain adaptation) setting when dealing with highly diverse source domains. Additionally, public CSI datasets are generally unsuitable for cross-domain analysis due to data scarcity within individual domains. This limitation can potentially be mitigated by incorporating multiple sources, which provides a larger training dataset and improves model performance in low-data scenarios. The main contributions of our work in this paper are summarized as follows:✓Multi-source M-FUDA outperforms all the baseline methods for most of the transfer learning tasks performed on three publicly available CSI datasets utilized for creating cross-user, cross-environment, and/or cross-atmospheric conditions using device-free HAR.✓The proposed model is applied to a multi-source unsupervised domain adaptation (MUDA) setup and contrasted against a single-source unsupervised domain adaptation (SUDA) setup designed with all the underlined sources combined in a single-source vs target setting. The proposed model produces promising results, surpassing traditional domain adaptation methods for device-free sensing. Our findings suggest that aligning multiple domain-invariant representations with domain-specific classifiers near class boundaries improves generalization. This alignment is particularly effective for each pair of source and target domains. As a result, the model performs well across various domain-shifting tasks.✓Empirical evaluation of various distance minimization approaches on one of the selected CSI datasets for each pair of source and target distributions indicates the suitability of maximum mean discrepancy (MMD) over others.✓Extensive evaluation shows that the predictive outputs of classifiers from different CSI sources capture target samples far from the support of underlying sources with the involvement of discrepancy and contrastive semantic alignment losses. This shows the role of proposed alignment losses in reducing the gap between the classifiers.

## 2. Related Work

Early reach work in device-free HAR has mainly been developed for a controlled laboratory environment without changing system characteristics from the training to the testing domain [36,37,38,39]. Although the accuracy of these systems is quite high, they cannot be implemented for a domain-shifting task without retraining the model with new target domain-labeled data samples. Without a doubt, is impractical to retrain models for every new domain, and sufficient labeled data for each new domain are not always available. Single-source and combined-source unsupervised domain adaptation (UDA) have been widely adopted in device-free HAR, confronting these shortcomings [27,40,41]. FewSense [42] is a few-shot learning (FSL) approach [43,44] that uses two methods for feature generation. The authors used a trained feature extractor to produce feature embeddings directly from a support set, called direct feature matrix generation. They also utilized an untrained classifier with the feature extractor and optimized their weight matrix using the support set, called fine-tuned feature matrix generation. Then, they computed the cosine similarity between the latent features of a query set and generated a feature matrix to train the model for novel classes introduced in cross-domains. WiGR [45] and DFGR [46] are also FSL-based deep similarity evaluation networks for activity classification in cross-domain Wi-Fi sensing. Features extracted from a query sample are concatenated with the features of every support sample, and their similarity is computed in a CNN-based similarity evaluation network using episode-based training. Network parameters are updated until the dissimilarity between support and query sets is small. Convolutional neural networks (CNNs) have shown remarkable contributions in the field of computer vision for image recognition tasks. CNN, despite its low preprocessing requirements, can explore the spatial and temporal dependencies in an image, which is also very helpful for HAR applications. Moshiri et al. [47] used a public raw CSI dataset [48] to generate grayscale CSI images using black-and-white colormaps. These images were fed into two 2D- convolutional layers of 3 × 3 filter size, each of them followed by a 2 × 2 max pooling layer, ReLU activation, and batch normalization. Finally, a dense neural network with softmax activation was used to produce class predictions. Model overall accuracy was 89.22% with 91% accuracy for fall-like in-domain activities, and it is thus suitable for healthcare applications. Moshiri et al. [49] produced pseudo-color RGB scale CSI images and fed them to a 2D-CNN model. Model overall accuracy was around 95% better than long short-term memory (LSTM) and bi-directional LSTM regarding training time and recognition accuracy. However, the model applicability was only tested for domain-specific tasks. DASAN [50] is an attempt to use pseudo-color RGB scale CSI images for the analysis of system sustainability in cross-user domain-shifting variations using inter-domain and intra-domain alignment techniques. Changsheng Zhang and Wanguo Jiao [51] performed five transformations on time series CSI data to convert them into images for the first time. These introduced CSI transformations were Gramian angular sum field (GASF) transformation [52], recurrence plot (RP) transformation [53], Gramian angular difference field (GADF) transformation [52], short-time Fourier transportation [54], and Markov transition field (MTF) transformation [52]. A three-layer 2D-CNN architecture was employed to recognize the activity from the converted images. Their results showed the superiority of RT over other techniques; however, this was determined through in-domain testing, which is most likely to achieve good results. The recognition accuracy of these CSI transformations in cross-domains is yet to be explored. Since CSI data streams vary in terms of time and correlate with target actions over a certain period, we can therefore assume them to be CSI images with temporal dependencies and leverage the extraordinary capabilities of CNNs for identifying such actions as an image recognition problem. There has been extensive research work in the last few years regarding the very same idea of representing CSI data streams in images and making the most of CNN-based architectures for device-free HAR. In the study of [55], a four-stage fall detection mechanism was presented. (1) They collected raw CSI data. (2) They denoised these data using discrete wavelet transform (DWT) [56]. DWT increased the signal-to-noise ratio and reduced the mean square error. (3) Short-time Fourier transformation [57] was applied to obtain CSI time-frequency diagrams for model training. Frequency domain analysis is effective for eliminating environmental influences and can identify the same activities performed at different venues. (4) A pre-trained GoogLeNet model [58] was further trained using time-frequency spectrograms for new domain classifications. Likewise, the work of [59] exploited the time-frequency spectrograms generated from a principle component analysis (PCA) of the first 20 impactful data features extracted from multiple sub-carriers. A generator and two asymmetric classifiers were operated in an adversarial fashion to maximize the discrepancy near the decision boundaries of the classifiers. Finally, the generator was optimized to produce target data features in support of source samples. This increased model accuracy in domain-shifting variations caused by unseen users and new locations.

Recent solutions for cross-domain Wi-Fi sensing are intended to align a single source domain to a single target domain or a multi-source combined domain to a single target domain. Fidora [60] is a wireless localization system designed to overcome Wi-Fi fingerprint inconsistencies caused by factors like body shape variations, background objects, and environmental changes. The novelty of their approach is the generation of augmented samples from collected CSI data using a data augmenter consisting of eight variational auto-encoders (VAE), one for each location. The augmented and original data were passed through feature extraction layers followed by classification and reconstruction layers. The classifier predicted location labels, while the reconstruction layers regenerated the original input samples. Compared to baseline models such as AutoFi [61], VAE-only, and FiDo [62], Fidora demonstrated a significant performance improvement, achieving 17.8% and 23.1% higher F1 scores in cross-user and cross-environment scenarios, respectively. WiGR [45] is a Wi-Fi-based few-shot learning system for gesture recognition, capable of domain adaptation across different environments. It employs supervised learning to generalize across new tasks using limited data. The CSI phase values, preprocessed using phase unwrapping and an FIR filter, were used for training. The model consists of a feature extraction subnetwork and a similarity discrimination subnetwork. The extracted features are combined, and similarity scores determine gesture classifications. WiGR outperformed competing models, including WiGeR [63], WiCatch [64], SignFi [65], and Siamese-LSTM [66], across different users, environments, and locations.

In our previous study [50], we examined the impact of transitioning users from the training phase to the testing phase on the predictive performance of an adversarial model named DASAN-MMD. This model utilized CSI data in a single-source-domain to single-target-domain setup. However, our proposed approach struggled to achieve high accuracy in some of the cross-user tasks due to the limited number of per-domain samples available in the public dataset [49]. Therefore, the combination of diverse source domains in a single common domain helps to enlarge the data distribution and confirms the effectiveness of SUDA methods in certain cases [27,41,59]; however, the improvement might not be significant or guaranteed in many domain-shifting tasks. This demands a better way to transform multi-source domains into a diverse target domain via multi-domain feature alignment at intermediate levels and near class boundaries. Our proposed model—Multi-Feature Unsupervised Domain Adaptation (M-FUDA)—makes full use of multiple source distributions and minimizes mismatch among diverse source and target domains in the first place. Second, it reduces the disparity among domain-specific classifiers at decision boundaries and increases the degree of accuracy using the average of multiple classifier outputs over the target domain. Finally, it diminishes the distance between samples of positive classes and increases the distance between samples of negative classes belonging to diverse source and target domains. Our model is a modified version of a previously published work [67] which has shown its superiority over other MUDA methods [68] in numerous domain shifting tasks due to its multi-level alignment strategy.

## 3. Preliminaries

### 3.1. Channel State Information (CSI)

The CSI represents detailed information about the state of a communication channel being impacted by various environmental causes during the signal transmission from the transmitter (TX) to the receiver (RX). The radio transmission is a multi-path propagation of Wi-Fi signals that travels through several paths facing line-of-sight (LOS) and non-line-of-sight (NLOS) contacts from TX to RX in an indoor environment. During a multi-path propagation, emerging signals have different time delays and attenuate differently with varying phase shifts due to reflection, scattering, diffraction, fading, and interference produced by the surrounding objects. In systems that support multiple-input multiple-output (MIMO) antennas, the CSI can efficiently use the spatial diversity of orthogonal frequency division multiplexing (OFDM) and can carry multiple copies of the same signal characterized by sub-carriers. The channel impulse response of the estimated CSI at the receiving end is defined as [33](1)$$H_{x,y,z}=\sum_{i}^{N}A_{i}\exp^{-j\times 2\pi \times d_{x,y,i}\times f_{z}/c}$$

Also,(2)$$H_{x,y,z}=\underset{magnitude}{\underset{︸}{|H_{x,y,z}|}}\exp^{-j\underset{phase-shift}{\underset{︸}{∠H_{x,y,z}}}}$$
where *N* is the total number of propagation paths, out of which the i-th path length from the x-th transmit antenna to the y-th receiver antenna is $d_{x,y,i}$, bearing an amplitude of $A_{i}$; $f_{z}$ is the z-th sub-carrier frequency; and *c* is the speed of radio waves in a vacuum, which is approximately $3\times 10^{8}$ m/s. Equation (Equation 1) is the simplified form of the estimated CSI, neglecting the impacts of cyclic shift diversity (CSD), sampling time offset (STO), sampling frequency offset (SFO) and beamforming due to signal modulation and demodulation; TX and RX hardware imperfections; and software errors.

When all the objects in an indoor environment are static, an occupant movement can be traced through a characteristic change in the CSI magnitude and phase-shift over multiple sub-carriers caused by the target movement. These characteristic variations create a principle background on how deep models are trained for device-free sensing. Suppose *a* is a transmitted radio signal. Upon reception, it is estimated to be b=H×a+n, where *n* is the noise vector, *b* is the received signal, and *H* is the CSI matrix. The estimated CSI is a 3-D matrix of complex values that variate along the time axis. The time axis is the fourth dimension that refers to the knowledge of the target movement with respect to time. In a Wi-Fi-based MIMO system having *x* transmit antennas and *y* receive antennas divided among *z* sub-carriers, the 4-D CSI matrix is denoted as(3)$$H_{x,y,z}={\left[MATRIX_{\left(CSI\right)}\right]}^{x\times y\times z\times t}$$
where *x* and *y* represent the spatial diversity, *z* shows the diversity in the frequency domain, and *t* is the time diversity in terms of the data packets sent to the receiver.

### 3.2. Wasserstein Distance

The Wasserstein distance measures the minimal cost required to match two dissimilar probability distributions by reshaping one into the other. Suppose *P* and *Q* are two different probability distributions in terms of mass over space. The minimal cost is the least amount of work needed to move mass *m* to a distance *d* in order to transform probability distribution *P* into probability distribution *Q*. It has gained popularity in domain shifting tasks due to its robust support for transferring knowledge from the source domain to the target domain even when the distributions are far apart. Mathematically, the measure of 1D-Wasserstein distance between probability distributions *P* and *Q* is defined as [69,70,71](4)$$W\left(P,Q\right)=\underset{\gamma \in \Gamma (P,Q)}{\inf}E_{(x,y)\;\gamma }[||x-y||]$$
where Γ(P,Q) is the set of all joint distributions with marginals *P* and *Q*, and ||•|| is the L1-norm. When *P* and *Q* are represented as finite discrete samples, say *u* and *v* with *n* samples each, the 1D-Wasserstein distance simplifies significantly, which is given by(5)$$W\left(P,Q\right)=\frac{1}{n}\sum_{i=1}^{n}\left|u_{i}-v_{i}\right|$$
where $u_{i}$ and $v_{i}$ are the sorted values of the samples *u* and *v* for pairing points optimally in terms of transportation cost in a single dimension. Sorting these samples mimics the effect of minimizing the transportation cost in uni-direction and computing their absolute differences, representing the transportation cost for each pair of matched points. Finally, taking the mean results in the average optimal transportation cost, which is the objective of the 1D-Wasserstein distance [71].

### 3.3. Correlation Alignment

Correlation alignment is another kind of statistical analysis for minimizing the difference between two covariance matrices acquired from the extracted features of the source and the target domains. Suppose the extracted source and target features are $X_{s}$ and $X_{t}$, respectively. These generated features are used to calculate pairwise correlations which are defined as [72](6)$$C_{s}=\frac{1}{n-1}\left(X_{s}^{T}X_{s}\right)-\frac{1}{n^{2}}\left(X_{s}^{T}1\right)\left(1^{T}X_{s}\right)$$(7)$$C_{t}=\frac{1}{n-1}\left(X_{t}^{T}X_{t}\right)-\frac{1}{n^{2}}\left(X_{t}^{T}1\right)\left(1^{T}X_{t}\right)$$
where 1 is a vector of ones and *n* is the number of data points. $C_{s}$ and $C_{t}$ are the covariance matrices generated from $X_{s}$ and $X_{t}$, respectively.

Finally, the element-wise difference between the covariance matrices $C_{s}$ and $C_{t}$ is computed to measure the CORAL loss, known as the Frobenius norm. CORAL loss is defined as [72,73](8)$$L_{CORAL}=\frac{1}{4d^{2}}\left|\right|C_{s}-C_{t}{\left|\right|}_{F}^{2}$$
where $\left|\right|•{\left|\right|}_{F}$ represents the Frobenius norm, and *d* is the dimensionality of the $X_{s}$/$X_{t}$.

### 3.4. Maximum Mean Discrepancy and Its Variants

Maximum mean discrepancy (MMD) is a statistical method that transforms the mean embeddings of two probability distributions into a high-dimensional reproducing kernel Hilbert space (RKHS). The RKHS is a special type of space where comparing two probability distributions is simpler. Let us suppose that we have two probability distributions *P* and *Q*. Their mean embeddings in the RKHS are defined as(9)$$\mu_{P}=E_{(x)\;P}\left[\phi \left(x\right)\right]$$(10)$$\mu_{Q}=E_{(x)\;Q}\left[\phi \left(x\right)\right]$$
where ϕ(•) is a mapping function which maps the features into the RKHS. ϕ(•) is associated with a characteristic kernel function $k\left(X_{s},X_{t}\right)=<\phi \left(X_{s}\right),\phi \left(X_{t}\right)>$, and $X_{s}$ and $X_{t}$ are the source and target domain features, respectively. In deep learning, MMD loss is computed in order to minimize the discrepancy between distributions *P* and *Q* by measuring the difference of the source and target mean embeddings in the RKHS [74]: (11)$$L_{MMD}\left(P,Q\right)=\left|\right|\mu_{P}-\mu_{Q}{\left|\right|}_{H}$$
where *H* is the reproducing kernel Hilbert space (RKHS). When the kernel function used in MMD is changed to a multi-kernel function ${\left\{k_{i}\left(X_{s},X_{t}\right)\right\}}_{i=1}^{M}$, it helps to leverage the strengths of multiple kernels to adapt to complex differences between source and target distributions. These kernels are useful for extracting features from different aspects of data using a balancing factor that decides the contribution of each aspect to the overall alignment of two distributions to achieve better adaptation. Such an alignment technique is called multiple-kernel maximum mean discrepancy (MK-MMD) [75,76]. Mathematically, a multi-kernel function is defined as(12)$$K\left(X_{s},X_{t}\right)=\sum_{i=1}^{M}W_{i}k_{i}\left(X_{s},X_{t}\right)$$
where $W_{i}$ is a balancing factor for which $W_{i}\geq 0$ and $\sum_{i=1}^{M}W_{i}=1$. MMD aligns a single feature between two probability distributions $P(X_{s})$ and $Q(X_{s})$ using a single kernel function $k(X_{s},X_{t})$ to compare samples. We map multiple feature spaces or joint feature representations of two probability distributions to a reproducing-kernel Hilbert space (RKHS) such that their joint probability distributions are $P(X_{s}^{1},X_{s}^{2},\ldots ,X_{s}^{M})$ and $Q(X_{t}^{1},X_{t}^{2},\ldots ,X_{t}^{M})$ where ${\left\{X_{s}^{j}\right\}}_{j=1}^{M}$ and ${\left\{X_{t}^{j}\right\}}_{j=1}^{M}$ are the source and target samples, respectively. A single-layer kernel is now replaced with a joint kernel of individual layers using $K(\left(X_{s}^{1},\ldots ,X_{s}^{M}\right),\left(X_{t}^{1},\ldots ,X_{t}^{M}\right))$, and such a variant of MMD is called a joint maximum mean discrepancy (JMMD) [77]. This mapping may involve more complex and powerful alignment; however, it is computationally heavier. We cannot always guarantee improvement in matching source and target distributions robustly using any of these techniques because they do not explicitly consider class labels or their relationships, especially in imbalanced data, as they may fail to align class-specific features effectively. Therefore, it is an exercise in trial and error to find their suitability in particular scenarios.

## 4. Problem Definition

Let us suppose that we have *N* different Wi-Fi CSI data source distributions such that $D=\{D_{1},D_{2},D_{3},\ldots ,D_{N}\}$, and their probability distributions are represented as ${\left\{P_{s_{i}}\left(X_{s_{i}},Y_{s_{i}}\right)\right\}}_{i=1}^{N}$. These source distributions have distinct features in terms of sensed environmental characteristics taken from *N* heterogeneous spaces for which source distributions are not uniform, i.e., $D_{1}\neq D_{2}\neq D_{3}\ldots \neq D_{N}$. Each source domain is equipped with labels in such a way that $T_{s}={\left\{\left(X_{s_{i}},Y_{s_{i}}\right)\right\}}_{i=1}^{N}$, where $X_{s_{i}}={\left\{x_{s_{i}}^{j}\right\}}_{j=1}^{M_{s}}$ denotes the j-th sample of the i-th source domain, having a total of $M_{s}$ data samples in the underlying source domain and $Y_{s_{i}}={\left\{y_{s_{i}}^{j}\right\}}_{j=1}^{M_{s}}$ is the corresponding ground-truth label. A transfer learning task is to match these diverse source distributions to a target distribution $P_{t}\left(X_{t},Y_{t}\right)$, with domain data $T_{t}=\left\{\left(X_{t},Y_{t}\right)\right\}$ where $X_{t}={\left\{x_{t}^{j}\right\}}_{j=1}^{M_{t}}$ having no ground-truth labels ${\left\{y_{t}^{j}\right\}}_{j=1}^{M_{t}}$. This target domain is distinct from all the source domains, i.e., $D_{t}\neq \left\{D_{1},D_{2},D_{3},\ldots ,D_{N}\right\}$. Suppose that these diverse source domains are mapped into a common feature space $D_{s}$ and align with the target domain $D_{t}$ using traditional domain generalization methods. Such adversarial learning would mainly focus on common domain-invariant representations coming from all domains, which is not an optimal solution to generalize these diverse domains because of systems’ inability to align myriad domain-invariant multi-source data features in a common feature space. This is likely to lead to poor performance in the domain-shifting Wi-Fi sensing tasks.

This problem definition leads to the research question addressed in this paper: how best to align wireless sensor data for multi-feature domain distributions continuously to a new domain with no label observations so that the system can maintain high accuracy regarding heterogeneous domain discrepancies. A reasonable solution is an unsupervised network with multiple subnetworks, each used for mapping a pair of source and target domains into specific feature spaces and matching their distributions. This would entail minimizing the distance between domain-specific feature spaces via measuring the difference between two probability distributions, then utilizing domain-specific classifiers to classify each domain label before finally aligning the domain-specific classifiers’ outputs for the target domain via domain discrepancy minimization techniques near the domain-specific decision boundary [67]. Domain discrepancy minimization refers to the process of reducing the differences between a source domain and a target domain, each belonging to distinct data distributions, to improve model generalization across them. Additionally, reverting to point-wise surrogates of the source and target distribution distances and similarities might be effective in multi-domain generalization problems [78]. The main objective of this paper is to apply multi-feature unsupervised domain adaptation techniques discussed above to cross unaligned domain-specific distributions in device-free human activity classification so that the model can leverage full benefits from a multi-source environment for an optimal solution of the problem, to investigate whether the multi-source unsupervised domain adaptation (MUDA) setup successfully matches diverse source distributions to a target domain, and to outmatch the classical single-source unsupervised domain adaptation (SUDA) methods used earlier as multi-source combined solutions.

## 5. Materials and Methods

### 5.1. Proposed Method

This paper applies an unsupervised wireless sensor alignment approach to multiple cross-domains affected by one or more types of variations caused by unknown users, unseen environments, and/or surrounding atmospheric conditions in human activity recognition using multi-feature unsupervised domain adaptation (M-FUDA), inspired by the work of [67]. M-FUDA is a four-stage alignment procedure intended to efficiently use the diverse source domain features in domain transferring tasks. In this section, we will first present an overview of the proposed approach and then describe the technical details of each component.

### 5.2. Overview

Prior research work in the field of device-free sensing is mainly focused on adversarial models trained on a multi-source combined data features to extract domain-invariant representations of all domains and map them onto a new target domain for matching cross-domains with inconsistency problems [27,41,59]. At first, it is difficult for an adversarial model to efficiently extract domain-invariant features coming from a diverse range of source domains, and one-on-one mapping of these domains in a common feature space aligned with the target domain is also not convenient. This leads to poor recognition accuracy in many domain shifting tasks. Researchers also employed pre-trained models for transforming domains in device-free HAR in order to minimize the computational cost and training time and also to deal with the training data scarcity issues. However, transfer learning-based approaches are not very effective in reducing the training data proportions, as investigated by the work of [79]. They achieved comparable results to the adversarial models with 500 training data samples using a pre-trained roaming model that was already trained between environments A and B in order to learn quickly between environments A and C. Again, common feature extraction from multiple source domains is not easier using transfer learning-based approaches, especially when the training data samples of a new domain are very limited. Therefore, our proposed model is a combination of a shared pre-trained model connected with many adversarially trained domain-specific sub-networks. The first stage uses the benefits of transfer learning to extract domain-invariant representations of different pairings of multiple sources with a target domain. The second stage uses domain adaptation techniques to align each pair of source and target domains into specific feature spaces and match their distributions. To the best of our knowledge, this is the first attempt to test multi-source domain adaptation with a transfer learning-based method to assess the domain inconsistency problem in human activity classification using device-free sensing.

### 5.3. Domain Invariant Feature Alignment

The first part of our proposed architecture is a common feature extractor. This sub-network is a shared framework for N mutually distinct source domains and an unseen target domain different in features from any of the source distributions. It helps the model to map common domain-invariant features from their original feature spaces into a common feature space. Suppose we have a batch of *L* samples ${\left\{x_{s_{i}}^{j}\right\}}_{j=1}^{L}$ belonging to the j-th source domain and a batch of *L* samples ${\left\{x_{t}\right\}}_{j=1}^{L}$ belongs to the target domain. After passing through the common feature extractor F(•) with the mapping parameter $\theta_{f}$, they get transformed into $Z_{{f_{i}}_{s}}^{j}=F\left({\left\{x_{s_{i}}^{j}\right\}}_{j=1}^{L};\theta_{f}\right)$ and $Z_{f_{t}}=F\left({\left\{x_{t}\right\}}_{j=1}^{L};\theta_{f}\right)$. However, it would take a long time for the sub-network to converge on a common feature space from a diverse range of source distributions. Therefore, we utilize the resources of transfer learning with the help of a pre-trained convolution neural network (CNN) model. The sub-network training starts with the fine-tuning of a ResNet50 [80] architecture pre-trained on the ImageNet 2012 dataset. We follow the fine-tuning steps explained by the study of [81]. ImageNet 2012, a part of the ImageNet project [82], contains 1000 classes of images including animals, everyday objects, tools, etc. A ResNet50 architecture pre-trained on 1.2 million such images belonging to the family of the ImageNet project is the shared sub-network for all the source and target domains. We further fine-tune the sub-model weights with the diverse multi-domain sources and target domain data samples to make the feature vectors of all the domains as similar as possible and minimize domain discrepancy. The input to the ResNet50 sub-network is 3D CSI pseudo-color map images/spectrograms of size 64×64×3 (height, width, RGB channels). Figure 2 shows the architectural details of ResNet50 for fine-tuning the shared sub-network weights.

### 5.4. Domain-Specific Feature Alignment

As mentioned earlier, it is not always possible for an adversarially trained model to significantly extract similar data features from various dissimilar source distributions. In order to tackle this problem, we further feed common domain-invariant representations received from the previous Stage 1 of the shared sub-network into N unshared CNN-based domain-specific feature extractors in a group of batches. Each domain-specific feature extractor is utilized to map a pair of source and target domains into a specific feature space by producing the target representations in support of the paired source domain. Suppose $H_{i}\left(•\right)$ is the i-th domain-specific feature extractor mapping function with a mapping parameter $\theta_{h_{i}}$ introduced with the j-th source- and target-generated common data features $Z_{{f_{i}}_{s}}^{j}$ and $Z_{f_{t}}$, respectively. The domain-specific feature extractor produces the domain-specific feature spaces such that $Z_{{h_{i}}_{s}}^{j}=H_{i}\left(Z_{{f_{i}}_{s}}^{j};\theta_{h_{i}}\right)$ and $Z_{{h_{i}}_{t}}=H_{i}\left(Z_{f_{t}};\theta_{h_{i}}\right)$. To achieve this goal, we test several statistical methods to minimize the distance between two probability distributions so that they can be correlated to each other as similarly as possible. We evaluated the strength of three well-known domain discrepancy reduction techniques in our device-free cross-domain HAR, namely maximum mean discrepancy (MMD) [74], multiple-kernel maximum mean discrepancy (MK-MMD) [75,76], and joint maximum mean discrepancy (JMMD) [77]. Finally, we choose the best one in terms of its suitability for the specific HAR application. We add these distance minimization losses to the backpropagation loss of the classifier output, which is the third stage of the proposed architecture. Figure 3 depicts the architectural details of domain-specific feature extractors connected with the Stage 1 shared sub-network, illustrated in Figure 2.

### 5.5. Domain-Specific Classifier Alignment

A domain-specific classifier is a classification model trained on a paired source domain with a target domain and optimized for classifying unseen samples from the target domain. It is specifically designed to handle domain shifts, ensuring good generalization to the target domain despite differences in data distribution. The adversarial models trained for shifting domains usually tend to produce a one-to-one mapping between two distinct feature spaces. However, this approach is effective for coarse-grained applications when the two feature vectors are not significantly different. As the considered domain-transforming tasks in our case are being applied to fine-grained variations where the subject, environment, and/or atmospheric conditions are fully changed from the source to the target domain, such models are not expected to be very efficient in achieving good recognition accuracy for fine-grained domain shifting tasks in device-free HAR. These significant variations may have impact near to the decision boundaries of the classifier output and can misclassify similar source and target data features owing to dissimilarities created near to the decision boundaries.Models trained to transform a single source domain into a target domain often struggle to classify distinct classes near decision boundaries due to overlapping feature distributions. Data points near these overlaps exhibit characteristics of both classes, making it difficult for the model to assign high-confidence predictions. In our case, the source samples come from a diverse distribution which is highly likely to suffer from overlapping feature distributions and uncertainties in classifying target classes near decision boundaries.

To address this, the third stage of our proposed model involves *N* domain-specific predictors. Each of them receives a pair of source and target domain-specific invariant data features from the Stage 2 output of *N* unshared domain-specific feature extractors. Each classifier uses a softmax layer to classify the probability distribution into *K* classes. We exploit the disagreement of multiple classifiers in order to minimize the discrepancy near to the decision boundaries. Suppose $C_{i}\left(•\right)$ is the i-th domain-specific classifier mapping function with a mapping parameter $\theta_{c_{i}}$ introduced with the Stage 2 paired j-th source and the target-generated domain-specific data features $Z_{{h_{i}}_{s}}^{j}$ and $Z_{{h_{i}}_{t}}$, respectively. The domain-specific classifier produces the probability distribution divided among *K* classes such that $Z_{{c_{i}}_{s}}^{j}=C_{i}\left(Z_{{h_{i}}_{s}}^{j};\theta_{c_{i}}\right)$ and $Z_{c_{t}}=C_{i}\left(Z_{{h_{i}}_{t}};\theta_{c_{i}}\right)$. Suppose there are such *N* classifiers trained on a pair of source and target domain-specific data features of *N* total such distinct source domains. In that case, they are more likely to produce disagreement in the classifiers’ outputs for the unseen target samples due to their mismatch near to the class boundaries. Since the classifiers are trained on different source domains paired with a target domain, they are supposed to create this disagreement intuitively. Our objective is to minimize the discrepancy among classifiers’ outputs as much as possible and then take the average of all the classifiers’ outputs as the model’s final outcome. This final prediction has more confidence due to its weightage taken from the mutual agreement of *N* aligned classifiers on the target domain. To minimize this discrepancy, we utilize discrepancy loss as the absolute values of the differences among all pairs of classifiers’ probabilistic outputs. Suppose $P_{c_{i1}}\left(y_{i1}|x_{i1}\right)$ and $P_{c_{i2}}\left(y_{i2}|x_{i2}\right)$ are the softmax layer probabilistic outputs for i1-th and i2-th classifiers of *K* total classes. Their discrepancy loss is defined as(13)$$L_{disc}=\frac{1}{N\times (N-1)}\sum_{i2=1}^{N-1}\sum_{i1=i2+1}^{N}\left\{\frac{1}{K}\sum_{c=1}^{K}\left[\right|P_{c_{i1}}\left(y_{i1}|x_{i1}\right)-P_{c_{i2}}\left(y_{i2}|x_{i2}\right)\left|\right]\right\}$$

The discrepancy loss is finally added with each classifier classification loss, which is calculated using negative log likelihood loss (NLLoss), defined as(14)$$NLLoss=\frac{1}{L}\sum_{i=1}^{L}\left\{\frac{1}{K}\sum_{c=1}^{K}\log\left(P_{c_{i}}\left[y_{i}|x_{i}\right]\right)\right\}$$
where *L* is the number of data samples divided among a group of batches.

Figure 4 depicts the architectural details of domain-specific classifiers connected with the corresponding Stage 2 unshared domain-specific feature extractors, illustrated in Figure 3. These domain-specific classifiers follow a straightforward feed-forward neural network architecture with three fully connected layers.

### 5.6. Contrastive Semantic Alignment

The Stage 4 alignment of our proposed model is the contrastive learning and separation of the semantic probability distributions of paired source and target data features near to the critical decision boundaries. This semantically aligned yet maximally separated mapping of the embedded subspace effectively generalizes distinct domains with an extremely low number of labeled target training samples, as demonstrated by the study of [78]. However, our proposed approach is based on an unsupervised adaptation technique that does not have target label observations. Therefore, our first step is to pre-label target samples. The previous three stages of alignment steps have improved our model by increasing confidence in target label predictions. We utilize this setup to generate pseudo-labels on target samples from each of the domain-specific classifiers and leverage them to minimize the intra-class discrepancy from their counterpart paired source samples so as to reduce the distance between the samples within the same classes of source and target domains. At the same time, we maximize their inter-class differences so as to push the samples belonging to the dissimilar classes of source and target domains apart. This two-way optimization of intra-class and inter-class discrepancies is helpful to improve the domain generalization process using contrastive semantic alignment loss, another type of distance minimization method that is added with the other losses of the model and jointly helps in the adaptation strategy. Finally, an average of the multiple classifiers’ outputs increases the model performance in the recognition task. Mathematically, we can define it as(15)$$L_{CSA}=\frac{1}{2}\left(1-Y\right){\left[D(x_{s_{i}}^{j},x_{t}^{j})\right]}^{2}+\frac{1}{2}\left(Y\right)\max{\left[0,margin-D(x_{s_{i}}^{j},x_{t}^{j})\right]}^{2}$$
where *D* is the pairwise distance between source and target embeddings, *Y* shows the similarity between the source label and the target pseudo-label with a value of 0, and in the case of dissimilar pairs, it acquires a value of 1, the margin is declared as 1, and if the distance between dissimilar pairs goes beyond this margin, the loss of these pairs become 0 to avoid unnecessary separation.

Figure 5 shows the proposed architecture with the four stage alignment losses combined together as the overall loss for backpropagation of the model. It shows the overall architecture of the proposed model, with the three blocks having been separately explained in their layout design in the previous discussions of Figure 2, Figure 3 and Figure 4. Additionally, it introduces all the domain generalization losses used at different stages of model training. Also, Algorithm 1 explains all the training steps of our presented model. The total loss of the proposed architecture after the four-stage alignment technique is formulated as(16)$$L_{total}=L_{classification}+\alpha L_{Distance\_Minimization}+\beta L_{discrepancy}+\gamma L_{Contrastive\_Alignment}$$
where α, β, and γ are weights used to emphasize different losses. We fixed γ=0.02 during the model training and exponentially decreased α and β using the formula $\frac{2}{1+\exp(-10*p)}-1$, which progressively schedules these weights with a gradual change in *p* as the model training progresses [83].    
**Algorithm 1:** Multi-Source M-FUDA Training**Data**:*N* labelled source domains ${\left\{D_{s_{i}}\right\}}_{i=1}^{N}={\left\{{\left\{x_{s_{i}}^{\left(j\right)},y_{s_{i}}^{\left(j\right)}\right\}}_{j=1}^{M_{s}}\right\}}_{i=1}^{N}$ and unlabelled target domain $D_{t}={\left\{x_{t}^{\left(j\right)}\right\}}_{j=1}^{M_{t}}$**Result**:Optimized pre-trained ResNet50 (F) [80], *N* trained domain-specific feature extractors ${\left\{H_{i1}\right\}}_{i1=1}^{N}$, and *N* trained domain-specific classifiers ${\left\{C_{i2}\right\}}_{i2=1}^{N}$
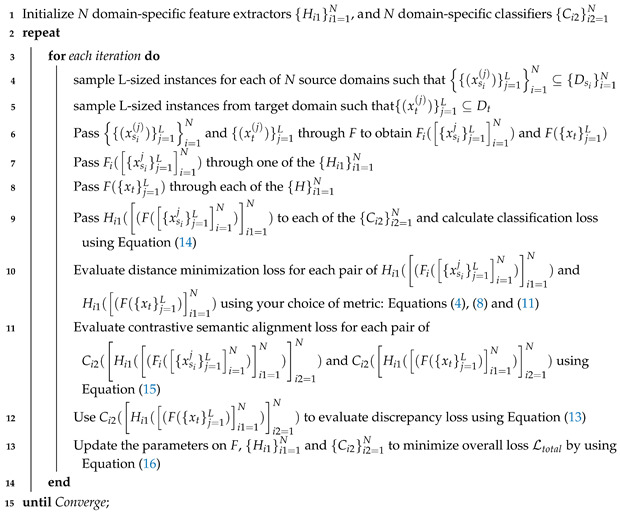


## 6. Experimental Results

The main objective of our evaluation is to assess the performance of M-FUDA in tackling the main types of domain adaptation tasks in human activity recognition using device-free sensing identified in the previous discussion as domain-shifting inconsistency produced due to varying users, environments, and/or atmospheric conditions from the training to the testing phase of the model. All experiments conducted in this study were performed on an Intel Core i7-9700K CPU (Intel Corp., Santa Clara, CA, USA) equipped with 32GB of internal memory and an Nvidia RTX 3060 GPU (Nvidia Corp, Santa Clara, CA, USA).

### 6.1. Datasets

We accessed three publicly available datasets [41,49,84] for our model evaluation on 24 September 2024. We have not had access to information that could identify individual participants during or after data collection. Regarding our first dataset, a personal computer using Raspberry Pi as a monitoring point (MP) and a TP-link archer c20 as an access point (AP) was employed to communicate over a single antenna pair with 52 useful sub-carriers, and a Nexmon Tool [85] was used to collect CSI data for seven daily activities repeated by 3 volunteers 20 times, each resulting in 420 samples in total. We used the magnitude values of the complex CSI to create cross-user domain transformations from the source to the target domain using the underlined dataset, which we called the Parisafm dataset [49]. For our assessment of cross-user and cross-environment variations overall, we utilized the Alsaify dataset [84], which was collected from 12 sets of activities repeated by 30 volunteers, with 20 trials for each, and 30 useful sub-carriers among 3 antenna pairs, resulting in 90 effective data features (columns) of complex CSI. Out of 12 sets of activities, we picked 6 activities performed by subjects 1, 2, and 3 in Environment 1; by subjects 11, 12, and 13 in Environment 2; and by subjects 21, 22, and 23 in Environment 3. Environments 1 and 2 were featured line-of-sight (LOS) contact between Tx and Rx, whereas Environment 3 presented a non-line-of-sight (NLOS) contact with the hurdle of a wooden wall of 8 cm between Tx and Rx. In total, we used 3240 samples of CSI magnitude values. For the Parisafm dataset, we repeated the preprocessing steps presented in [49] to produce 3D CSI images, whereas the preprocessing of the Alsaify dataset started with the normalization of data values between [0–255], applying Hampel filtering followed by principle component analysis (PCA) to extract 80 impactful data features. These PCA components were then employed to visualize t-distributed stochastic neighbor embedding (t-SNE) into 3 component dimensions. Finally, we generated CSI spectrograms from t-SNE output embeddings using short-time Fourier transform (STFT) [86] at a sampling rate of 200. For our last and very naturally occurring domain shifting task of varying atmospheric conditions from the training to the testing phase, we employed the dataset we called the Brinke and Meratnia dataset [41]. A mini-PC with an Intel Ultimate Wi-Fi Link 5300 NIC as a monitoring point and an access point (TP-LINK AC1750) were used to communicate over a 3×3 MIMO antenna pair with 30 useful sub-carriers, leading to 180 effective data features (columns), and a CSI Tool [87] was installed to collect CSI data for 6 daily activities repeated by 9 different participants over 6 multiple days. For our assessment, we extracted the data samples for 9 experiments of 4 activities performed by 2 participants over 3 days, and each activity was recorded for 50 trials of each experiment, generating 5400 samples in total. We only utilized CSI-magnitude for these samples, and we replicated the preprocessing steps explained in the study of [49]. Finally, we got a 3D-CSI image using MATLAB (2024) pseudocolor plots. Figure 6 presents the t-SNE plots of all three datasets with underlined proportions of samples mentioned in Table 1. These plots show the divergence and heterogeneity of samples in different classes coming from a diverse range of users, environments, and varying atmospheric conditions. This also demonstrates the difficulty level of domain adaptation tasks in the case of cross-domain alignments. The densely clustered data highlights the complexity of domain transformation. In our cross-user experiments, we worked with seven activities from the Parisafm dataset [49]. For our cross-user and cross-environment experiments, we used six activities from the Alsaify dataset [84] and four activities from the Brinke and Meratnia dataset [41]. These activities are labeled in Table 1, and each color in Figure 6 corresponds to an activity label in Table 1.

### 6.2. Configuration and Hyperparameter Tuning

Inspired by the foundational work of [67], we initiated our model simulations using the same hyperparameters as those employed in their study in order to establish a consistent baseline for comparison. The architecture begins with a shared sub-network based on a pre-trained ResNet50 [80], which outputs a feature space of size (1,1,2408). This shared sub-network leverages the robustness of ResNet50, trained on the ImageNet 2012 dataset, to extract generalized features applicable across different domains.

#### 6.2.1. Domain-Specific Feature Extractors

For each domain-specific task, the feature extractor comprises a combination of convolutional layers. Specifically, it includes two 2D−CNN(1×1) layers and one 2D−CNN(3×3) layer, followed by an average pooling layer of size (7,7). This configuration produces a refined feature space of size (1,1,2042), as depicted in Figure 3. The inclusion of 1×1 convolutions helps with dimensionality reduction and fine-grained feature selection, while the 3×3 convolutional layer captures spatial relationships and patterns critical for domain-specific tasks.

#### 6.2.2. Domain-Specific Classifiers

Each domain-specific classifier is implemented as a three-layer fully connected network, structured as 2042→256→K classes, where *K* represents the number of output classes for the specific domain, as depicted in Figure 4. A softmax activation function is applied to the final layer to output class probabilities. This structure ensures sufficient capacity for complex classification tasks while maintaining computational efficiency.

#### 6.2.3. Grid Search for Optimal Configuration

To enhance model performance, we conducted a thorough grid search by varying key architectural components. These variations included modifying the number of output channels and kernel sizes in the 2D−CNN layers as well as experimenting with deeper fully connected layers in the domain-specific classifiers. We also tested different batch sizes (10,30,60) during training. Interestingly, despite extensive experimentation, the most optimal configuration remained aligned with the one proposed in [67]. The only significant improvement was achieved by replacing the cross-entropy loss function with negative log-likelihood loss (NLLoss), which provided a noticeable boost in classification accuracy.

#### 6.2.4. Learning Rate Strategy and Optimization

Since the shared sub-network was pre-trained on ImageNet, its learning rate was set 10 times smaller than those for the domain-specific feature extractors and classifiers in order to preserve learned representations while allowing for fine-tuning. We tested learning rates in the range of [0.0001,0.01]. To further optimize training, we adopted a dynamic learning rate adjustment strategy based on [67]. The learning rate was updated after each epoch using the following formula: (17)$$LR_{new}=\frac{LR_{prev}}{{\left(1+10*p\right)}^{0.75}}$$
where *p* is a parameter that progresses linearly from 0 to 1 over the course of training, allowing for gradual refinement of the learning process, and 0.75 is set by the strategy design.

#### 6.2.5. Optimization Algorithms

We evaluated several optimization algorithms, including

Adaptive Moment Estimation (Adam): Known for its adaptive learning rate capabilities.Adaptive Gradient Algorithm (Adagrad): Effective for sparse data scenarios.Adam with Weight Decay (AdamW): Combines Adam’s efficiency with weight decay for better regularization.Stochastic Gradient Descent (SGD) with momentum (0.9): Provides stability and faster convergence by dampening oscillations.

Among these, SGD with momentum yielded the best performance, balancing optimization speed and model generalization.

#### 6.2.6. Early Stopping and Data Splits

To prevent overfitting and ensure robust model training, we implemented early stopping. The model’s overall loss was monitored after an initial warm-up phase, and training was halted if the loss did not improve over five consecutive epochs. This strategy helped maintain computational efficiency while optimizing model performance. The dataset was split into 60% for training, 20% for validation, and 20% for testing. This division ensured a fair evaluation of the model’s generalization capabilities. The final model was selected based on its high predictive performance on the validation set, particularly for domain-shifting tasks that challenged the adaptability of the architecture.

Our analysis revealed that while the original configuration proposed in [67] was highly effective, certain refinements—such as the adoption of NLLoss and fine-tuned learning rate strategies—led to further improvements. These modifications highlight the importance of nuanced adjustments in achieving state-of-the-art results for domain-adaptive learning tasks.

### 6.3. Comparison Techniques and Evaluation Metrics

Our proposed model consists of three blocks with four aligning stages. Stage 1 is a pre-trained model optimizing its weights on a multi-source CSI data distribution, producing a shared feature subspace for Stage 2 of *N* unshared domain-specific feature extractors. These feature extractors generalize the target distribution with each diverse source distribution using distance minimization approaches such as maximum mean discrepancy (MMD) [74]. We also tested two variants of maximum mean discrepancy (MMD) [74], namely multiple-kernel maximum mean discrepancy (MK-MMD) [75,76] and joint maximum mean discrepancy (JMMD) [77].

We first explored the sole impact of discrepancy and contrastive semantic alignment losses on the Parisafm dataset and then added these losses to one of the selected distance minimization losses. Finally, we chose the two best models in terms of high predictive performance and moderate simulation time and compared the state-of-the-art baseline adaptation techniques to the preferred versions of the recommended setup. For a fair comparison, all of the baseline methods followed the same deep learning architecture as the recommended version and were fine-tuned from a pytorch-provided model of ResNet50 [80]. The entire training process was repeated ten times for each domain-shifting task. Each time performance was evaluated on unseen target samples in order to take the average of these multiple trials for unbiased and satisfactory analysis. Our first baseline model was a combination of the source version of the proposed model with recommended model settings and MMD loss because this loss function successfully adopted the target domain with the highest or second-highest predictive performance among all of our tested multi-domain unsupervised domain adaptation (MUDA) techniques and did so with moderate simulation time. We used a single feature extractor and a classifier followed by the pre-trained ResNet50 [80] architecture and combined all the source domains into a traditional single-source vs target setting to testify to the importance of multiple sources being combined in training a prototype that has originated from a single unsupervised domain adaptation (SUDA) technique. Our second baseline model was derived from Wi-Adaptor [59], although we changed its architecture to match the proposed one in order to achieve a fair model comparison. Wi-Adaptor [59] trains a generator and two classifiers on combined-source samples to generate discriminative features and then utilizes the trained classifiers to maximize the discrepancy on target samples. Finally, it trains the generator on target samples to minimize the discrepancy. The authors’ contribution was inspired by the study of maximum classifier discrepancy (MCD) for unsupervised domain adaptation [88]. We further tested our proposed architectures against two classical distance minimization approaches—correlation alignment (CA) [72,73] and Wasserstein distance [69,70]—which have proven their effectiveness in domain adaptation tasks in earlier research works [72,89]. For these two baseline methods, we used the same architecture as in single-source M-FUDA, with the only difference being that the maximum mean discrepancy (MMD) loss after the domain-specific feature extractor was replaced by these distance minimization techniques. For better clarity, refer to Figure 5. We reported micro- and macro-F1 scores to assess the performance of different models. The best results for each specific task were colored in light green, and the best average of all the tasks was colored in light blue. Micro-F1 reports the models’ statistics based on the overall performance of the model, treating every instance as having equal importance, which is preferable in our imbalanced data distribution. Meanwhile, macro-F1 treats minority classes with the same importance as majority classes and reports the model performance across all the classes based on the weighted average of F1 scores for individual classes. Each table in our reported results presents the average of individual micro- and macro-F1 scores, calculated for each domain-shifting task performed on the selected datasets and displayed at the end of the table.

## 7. Results and Discussion

This section presents and discusses our results in three cross-domain experiments performed on three selected datasets. All experiments conducted in this study were performed on an Intel Core i7-9700K CPU equipped with 32GB of internal memory and an Nvidia RTX 3060 GPU.

### 7.1. Experiments with Varying Users

Our first experiment was to assess the model accuracy across different users to test how well it was able to generalize various users’ activities, each having different physical characteristics, and not become overfitted on any specific user in a pool of diverse source distributions. For this analysis, we chose the Parisafm dataset, which contains data from three users performing seven activities in a laboratory environment; see Table 1. This dataset was suitable for the underlined experiments because we were able to train our model on the features of any two users to investigate the learning adaptability of the third user as a target domain. We utilized three dissimilar source domains. Two of them belonged to single-source subjects 1, 2, or 3, and the third source contained combined data features of previously chosen subjects to achieve better commonality between diverse sources and target distributions. We first examined the significance of different alignment losses at various stages of neural network architecture. This started with the discrepancy (disc) loss, then we added contrastive semantic alignment (CSA) loss and included numerous distance minimization losses as we moved from left to right in Table 2 and Table 3. We finally deciphered the recommended settings for our proposed model, with discrepancy loss for aligning multiple classifiers’ outputs, contrastive semantic alignment loss for reducing the distance between samples of the same classes, and maximum mean discrepancy (MMD)/multiple-kernel maximum mean discrepancy (MK-MMD) loss for mapping target distribution onto the paired source domain. We called this selected model Multi-Feature Unsupervised Domain Adaptation (M-FUDA), which we employed for the first time for cross-domain applications in device-free HAR as our main contribution in this paper. Table 2 and Table 3 demonstrate our concluding remarks in terms of the superiority of the recommended settings over all the other combinations of loss functions and shows how this model outperforms other settings in three cross-user tasks in a multi-source environment. A multi-source environment consists of diverse source domains with related, yet distinct, data features. We leveraged these varied domains to expand the training dataset, addressing the limited availability of data within each domain of public datasets. Figure 7 illustrates a comparison of the different variants of multi-source M-FUDA, showcasing their respective performances under various loss functions.

In Table 4 and Table 5, we compared variants of multi-source M-FUDA to its combined-source version with MMD loss, CORAL [72], Wasserstein [89], and Wi-Adaptor [59], also known as MCD [88], trained on M-FUDA configurations taken for defining feature extractor and classifier architectures. We also used the same pre-trained model ResNet50 [80] as the initial block for all the baselines and fed them with combined source domains into a traditional single-source vs target setting to explore whether the multiple sources were valuable to exploit in a single-source domain adaptation (SUDA) setting or in the proposed M-FUDA in a multiple-source domain adaptation (MUDA) setting applied to cross unaligned domain-specific distributions in device-free human activity classification. The experimental micro- and macro-F1 results reported in Table 4 and Table 5 show that M-FUDA (MMD) is 13.78% better in terms of averaged micro-F1 score and 15.47% better in terms of averaged macro-F1 score than the combined-source MCD considering 3 domain shifting tasks created by 3 different users. Each task is repeated for 10 trials and an average figure is reported. These readings further validate that multi-source M-FUDA (MMD) is 3.85% higher in averaged micro-F1 score and 4.13% higher in averaged macro-F1 score than that of combined-source M-FUDA (MMD) calculated for 3 cross-user tasks run for 10 trials each. This substantiates that M-FUDA (MMD) can achieve a steadier recognition accuracy in a MUDA setting than its counterpart combined-source SUDA setups for cross-user domain shifting tasks in device-free HAR.

Similarly, M-FUDA (MMD) achieved very high recognition accuracy compared to CORAL [72] and Wasserstein [89], which ranked second-last and last in predictive performance, respectively. In contrast, M-FUDA (MK-MMD) outperformed M-FUDA (MMD) by 2.47% in averaged micro-F1 and macro-F1 scores. However, this improvement came at the cost of longer training times due to the higher computational complexity of multi-kernel evaluation, as shown in Table 10. It is evident that there was a big difference in the performance of multi-source M-FUDA vs combined-source SUDA methods, validating the superiority of the proposed architecture in cross-user conditions using device-free Wi-Fi CSI data. A detailed analysis of various cross-user tasks on the Parisafm dataset is presented below in which we compare the averaged micro- and macro-F1 scores of multi-source M-FUDA to baseline models, as illustrated in Figure 8.

### 7.2. Experiments with Varying Users and Environments

Our second experiment was designed to train our multi-source M-FUDA model for activities performed by users with different physical characteristics. Their performance also depended upon the environmental settings, such as the position of furniture, doors, windows, and transmission contact between the transmitter (Tx) and the receiver (Rx). The training data were not good enough to capture the diversity of the users and environments when the trained model was tested on varying external factors, for instance unseen users in unknown environments. Therefore, the transfer learning task was to assess our trained model’s robustness, adaptability, and transferability in terms of how well it generalized in such critical but very near to real-world scenarios. We utilized the Alsaify dataset to evaluate our model in such domain-changing applications using nine different users executing six different activities in three environments specified as (1) Jordan University of Science and Technology, a research laboratory (4.7 m × 4.7 m) area equipped with a Tx–Rx pair 3.7 m apart with a line-of-sight contact; (2) Jordan University of Science and Technology, Department, a university hallway (7.95 m × 3.6 m) area equipped with a Tx–Rx pair 7.6 m apart with a line-of-sight contact; and (3) Jordan University of Science and Technology, a room attached to a corridor fitted with a Tx inside the room and an Rx is outside the room with an 8 cm-thick wooden wall as a barrier causing a non-line-of-sight contact between the Tx–Rx pair, which are spaced 5.44 m apart. We also compared our model’s performance with baselines to investigate the impacts of model configurations and arrangements in multi-source MUDA vs combined-source SUDA settings. Multi-source M-FUDA (MMD) achieved the best micro- and macro-F1 scores on 7 out of 11 tasks when we changed both users and environments from the training to the testing phase, and this was even better than multi-source M-FUDA (MK-MMD) in terms of higher accuracy and shorter simulation time, as reported in Section 7.4. Our model obtained the highest averaged micro- and macro-F1 scores—75.6% and 75.0%, which is 3.70% and 3.59% higher than combined-source MCD [59,88], the second best performing technique, as reported in Table 6 and Table 7. It is also worth noting that the MUDA setting with the proposed alignments was able to achieve more balanced accuracy on individual instances and also per each class, leading to much higher micro- and macro-F1 scores than all the comparison techniques. In addition, multi-source MUDA (MMD) outperformed single-source M-FUDA (MMD) by 6.48% and 6.53% in micro- and macro-F1 scores; see Table 6 and Table 7. Performance comparison with multi-source M-FUDA (MMD) also showed superiority over combined-source CORAL [72] and Wasserstein [89].This demonstrates the effectiveness of utilizing diverse sources in the MUDA setting and the proposed architecture’s suitability for cross-user along with cross-environment domain changing setups in terms of capturing discriminative transfer features between classes using device-free Wi-Fi CSI data. A closer examination of various cross-user and cross-environment tasks on the Alsaify dataset revealed a comparison of the averaged micro- and macro-F1 scores for multi-source M-FUDA and the baseline models, as shown in Figure 9.

### 7.3. Experiments with Varying Atmospheric Conditions

Our last experiment was designed to train our multi-source M-FUDA model for activities performed by users on different days of the week. Since radio signals are sensitive to atmospheric conditions such as temperature, humidity, sunlight, darkness, wind speed, visibility, etc., the model’s adaptability to these typical scenarios is also an essential factor in analyzing the recognition accuracy in MUDA vs SUDA settings. To the best of our knowledge, we are the pioneers in assessing the performance of an adaptation model in cross-atmospheric conditions with the help of device-free Wi-Fi sensing. We found the Brinke and Meratnia dataset suitable for this assessment because we could use the CSI data for two participants performing activities on three different days of a week marked as day6, day7, and day8 in the public dataset. We focused on four activities, including clapping, falling, nothing, and walking. Multi-source M-FUDA (MMD) achieved the best micro-F1 and macro-F1 scores on 5 out of 6 tasks when we switched between day6, day7, and day8 from the training to the testing phase. It obtained the highest averaged micro- and macro-F1 scores—73.3% and 73.1%, which was 1.38% and 1.39% higher than combined-source M-FUDA (MMD) and MCD [59,88], as reported in Table 8 and Table 9. The proposed multi-source M-FUDA (MMD) also successfully outperformed multi-source M-FUDA (MK-MMD) and combined-source CORAL [72] and Wasserstein [89]. However, there was not a big difference in the predictive performance of multi-source M-FUDA vs combined-source SUDA methods in cross-atmospheric conditions, showing the limitations of the proposed architecture. M-FUDA is a computationally expensive architecture because we need to train each pair of source and target domains separately, and increasing the diversity in domains demands high computational costs, which is not advisable for minor accuracy improvements in scenarios such as cross-atmospheric conditions domain-shifting tasks, as demonstrated in Table 10. The Brinke and Meratnia dataset was analyzed for cross-atmospheric tasks, comparing the averaged micro- and macro-F1 scores of multi-source M-FUDA with those of baseline models, as depicted in Figure 10. Last, we compared the training times of the proposed M-FUDA variants with baseline methods across cross-domain tasks, as illustrated in Figure 11.

### 7.4. Computational Complexity

Table 10 reports the average computational time for all cross-user, cross-user with cross-environment, and cross-atmospheric experiments performed in Table 4, Table 5, Table 6, Table 7, Table 8 and Table 9 for the proposed multi-source M-FUDA and baseline methods. It is evident that M-FUDA is a complex model architecture, as it trains domain-specific feature extractors and classifiers separately for each pair of source and target domains. Consequently, incorporating more diverse sources increases model complexity and computational cost. However, this approach can also enhance predictive performance, as the model learns separately from diverse source domains. Thus, the proposed model presents a trade-off between improved recognition accuracy and higher computational cost.

To mitigate complexity, our model adopts a relatively simple architecture, utilizing a pre-trained ResNet50 [80] block while maintaining a minimal design for other blocks with only a few layers. This deliberate simplification helps reduce model complexity. However, despite these efforts, the simulation time remains higher compared to single-source SUDA methods. This highlights a limitation of our proposed model in scenarios where the predictive performance does not significantly justify the increased computational cost.

## 8. Conclusions

This paper introduces multi-source M-FUDA, a technique for unsupervised domain adaptation (UDA) applied to Wi-Fi-based human activity recognition. Unlike traditional single-source UDA methods, M-FUDA integrates multiple source domains, leading to better recognition accuracy and robustness. The model uses a four-stage alignment process with a pre-trained model, domain-specific feature extractors, and classifiers, along with a maximum mean discrepancy (MMD) loss for better performance and simpler architecture.

The approach minimizes the discrepancy between classifiers for target samples and uses contrastive semantic alignment loss to improve classification across source–target domain pairs. Evaluations on three public datasets show that M-FUDA outperforms single-source UDA in scenarios where domain samples belong to diverse distributions. However, the model’s higher complexity introduces a trade-off between computational cost and performance, limiting its applicability in low-gain, high-cost situations.

Future work will explore the model’s effectiveness in more challenging scenarios, such as multi-target tracking, using more complex architectures to improve predictive accuracy.

## Figures and Tables

**Figure 1 sensors-25-01876-f001:**
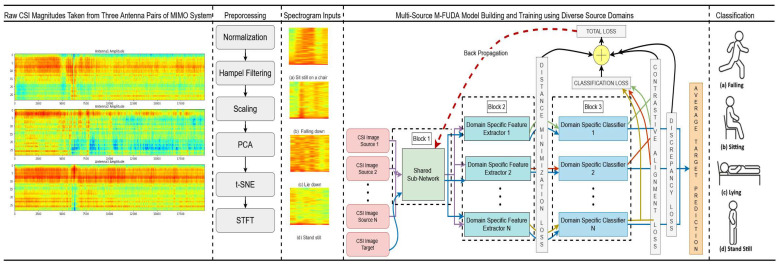
System overflow. Dark red indicates higher amplitude; blue indicates moderate amplitude; yellow and green indicate lower amplitude.

**Figure 2 sensors-25-01876-f002:**
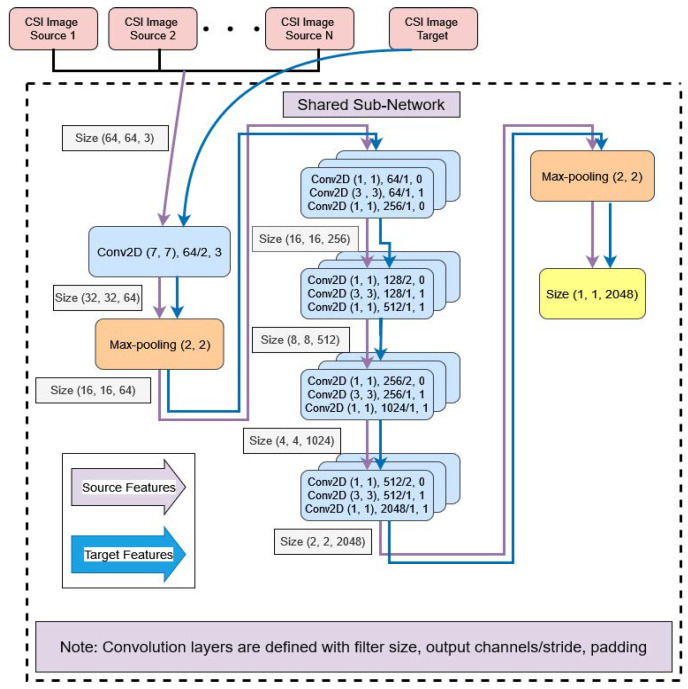
Architectural details of ResNet50 for fine-tuning the shared sub-network.

**Figure 3 sensors-25-01876-f003:**
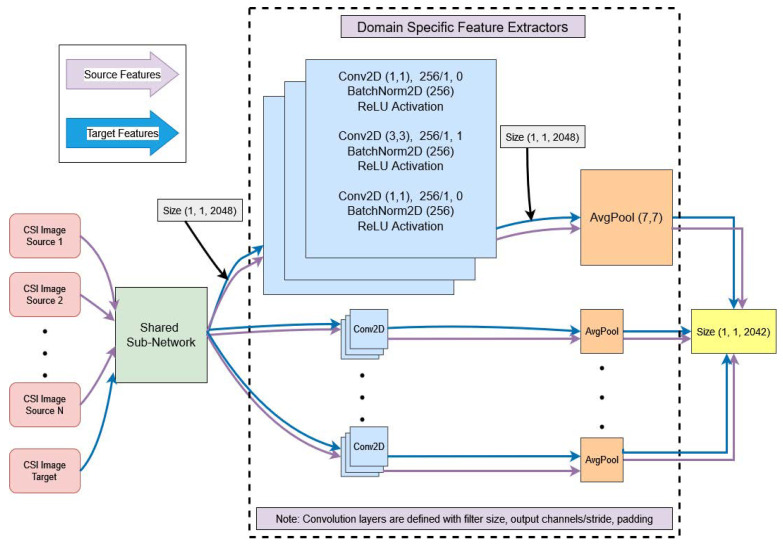
Architectural details of domain-specific feature extractors.

**Figure 4 sensors-25-01876-f004:**
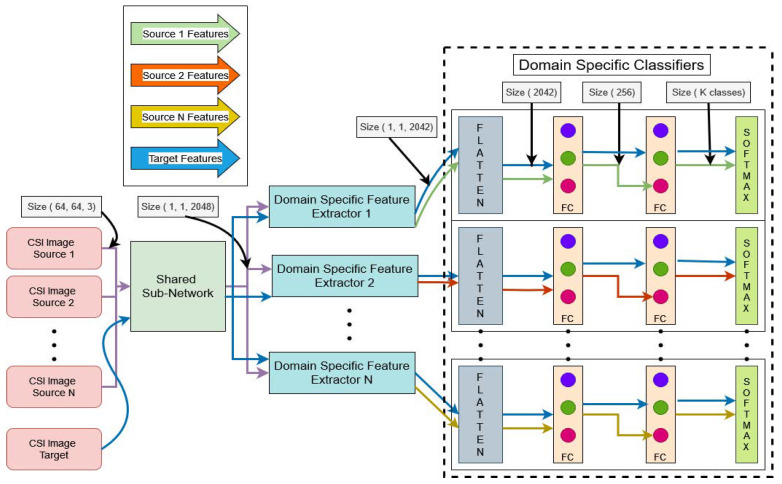
Architectural details of domain-specific classifiers.

**Figure 5 sensors-25-01876-f005:**
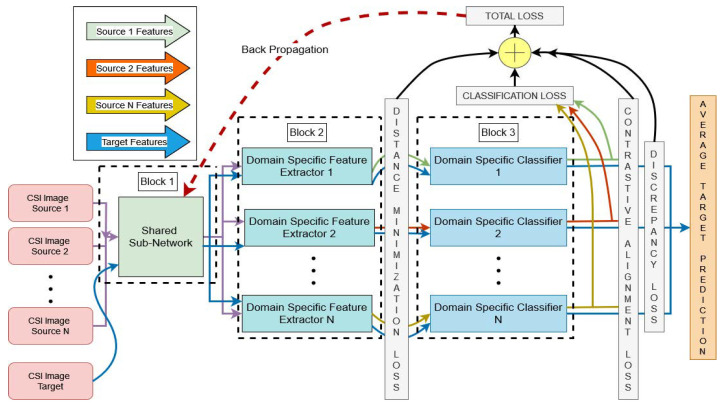
Model proposed architecture with four-stage alignment losses.

**Figure 6 sensors-25-01876-f006:**
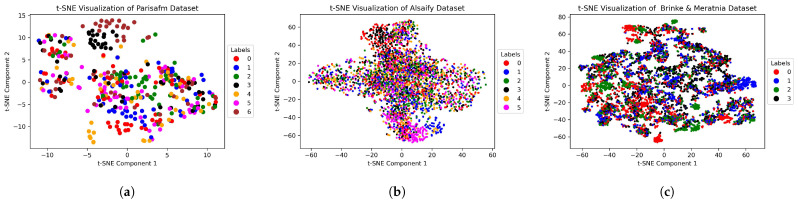
t-SNE plots for the selected datasets. The activity labels are shown in Table 1. (**a**) t-SNE visualization of the Parisafm dataset. (**b**) t-SNE visualization of the Alsaify dataset. (**c**) t-SNE visualization of the Brinke and Meratnia dataset.

**Figure 7 sensors-25-01876-f007:**
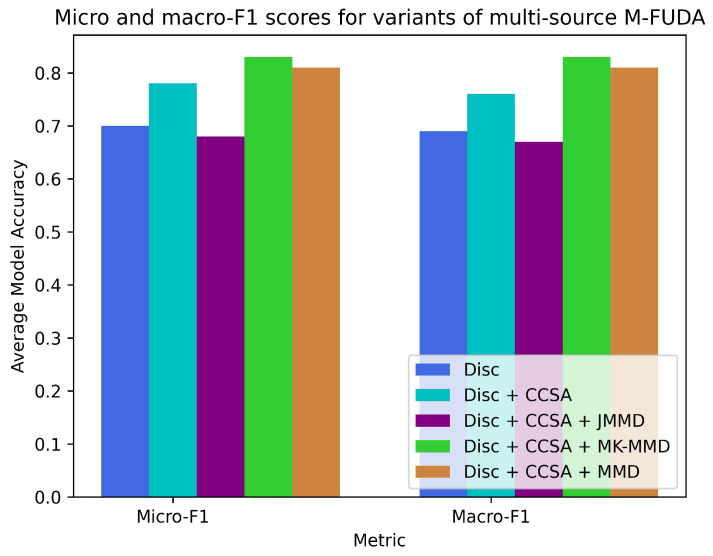
Comparison of average micro- and macro-F1 scores for variants of multi-source M-FUDA across different cross-user tasks.

**Figure 8 sensors-25-01876-f008:**
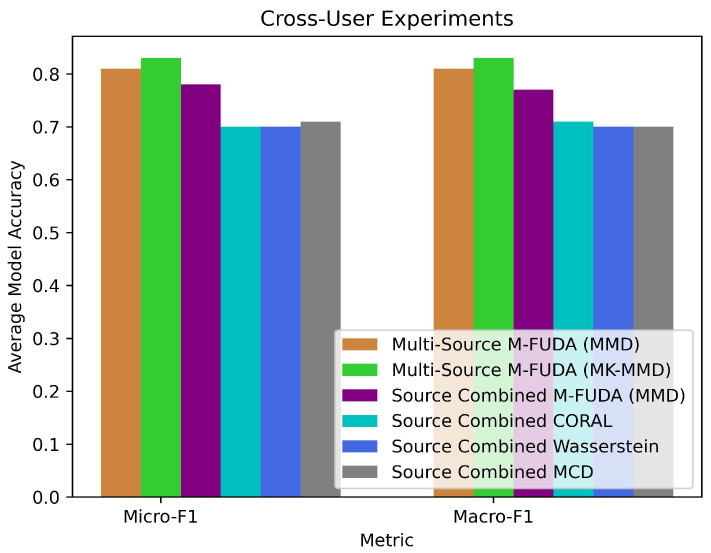
Comparison of average micro- and macro-F1 scores for multi-source M-FUDA and baseline models across cross-user tasks.

**Figure 9 sensors-25-01876-f009:**
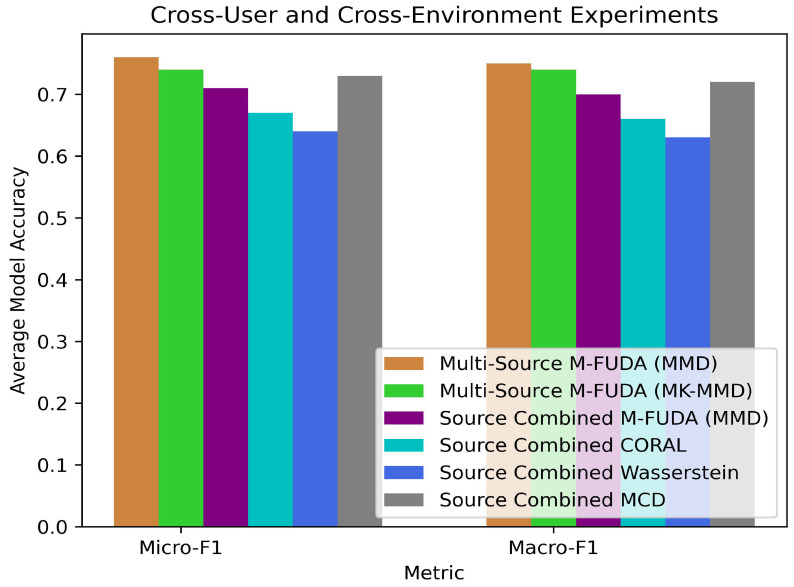
Comparison of average micro- and macro-F1 scores for multi-source M-FUDA and baseline models across cross-user and cross-environment tasks.

**Figure 10 sensors-25-01876-f010:**
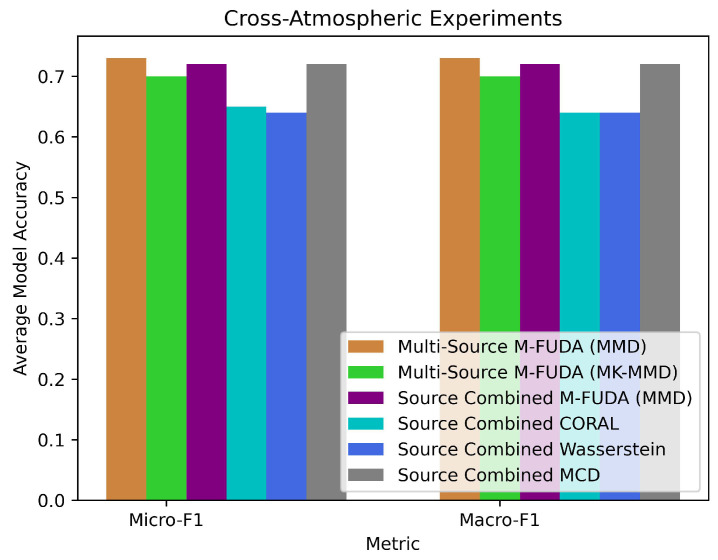
Comparison of average micro- and macro-F1 scores for multi-source M-FUDA and baseline models across cross-atmospheric tasks.

**Figure 11 sensors-25-01876-f011:**
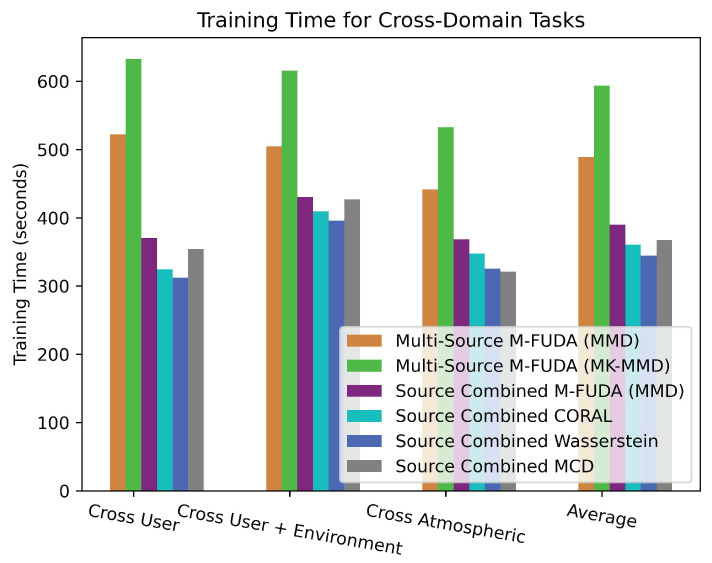
Comparison of training time for multi-source M-FUDA and baseline models across cross-domain tasks.

**Table 1 sensors-25-01876-t001:** Descriptions of datasets.

Dataset	No. of Features	No. of Samples	Antenna Pairs	No. of Users	No. of Environments	Atmospheric Impacts	Activities
Parisafm	52	420	1	3	1	Disregarded	(0) bending, (1) falling, (2) lie down, (3) running, (4) sit down, (5) stand up, and (6) walking
Alsaify	90	3240	3	6	3	Disregarded	(0) sit still on a chair, (1) falling down, (2) lie down, (3) stand still, (4) walking from the transmitter to the receiver, and (5) pick a pen from the ground
Brinke and Meratnia	270	5400	6	2	1	Considered	(0) clapping, (1) falling, (2) nothing, (3) walking

**Table 2 sensors-25-01876-t002:** Micro-F1 scores for variants of multi-source M-FUDA. Green background indicates the best accuracy for an individual task; blue background indicates best average accuracy; yellow background and bold font highlight the average performances.

Micro-F1								
Source Domains			Target Domain	Proposed Methods				
				Variants of Multi-Source M-FUDA				
Sourrce 1	Source 2	Source 3	Target 1	(Disc)	(Disc + CCSA)	(Disc + CCSA + JMMD)	(Disc + CCSA + MK-MMD)	(Disc + CCSA + MMD)
**S1**	**S2**	**S1 + S2**	**S3**	0.74	0.85	0.69	0.86	0.86
**S2**	**S3**	**S2 + S3**	**S1**	0.62	0.67	0.67	0.83	0.73
**S1**	**S3**	**S1 + S3**	**S2**	0.75	0.84	0.69	0.79	0.84
**Average**				**0.70**	**0.78**	**0.68**	**0.83**	**0.81**

**Table 3 sensors-25-01876-t003:** Macro-F1 scores for variants of multi-source M-FUDA. Green background indicates the best accuracy for an individual task; blue background indicates best average accuracy; yellow background and bold font highlight the average performances.

Macro-F1								
Source Domains			Target Domain	Proposed Methods				
				Variants of Multi-Source M-FUDA				
Sourrce 1	Source 2	Source 3	Target 1	(Disc)	(Disc + CCSA)	(Disc + CCSA + JMMD)	(Disc + CCSA + MK-MMD)	(Disc + CCSA + MMD)
**S1**	**S2**	**S1 + S2**	**S3**	0.73	0.81	0.66	0.86	0.83
**S2**	**S3**	**S2 + S3**	**S1**	0.61	0.65	0.66	0.82	0.74
**S1**	**S3**	**S1 + S3**	**S2**	0.75	0.83	0.70	0.80	0.85
**Average**				**0.69**	**0.76**	**0.67**	**0.83**	**0.81**

**Table 4 sensors-25-01876-t004:** Micro-F1 scores for multi-source M-FUDA and baseline models across cross-user tasks. Green background indicates the best accuracy for an individual task; blue background indicates best average accuracy; yellow background and bold font highlight the average performances.

Micro-F1									
Source Domains			Target Domain	Proposed Multi-Source Models		Combined-Source Models			
Source 1	Source 2	Source 3	Target 1	M-FUDA (MMD)	M-FUDA (MK-MMD)	M-FUDA (MMD)	CORAL [72]	Wasserstein [89]	MCD [59,88]
**S1**	**S2**	**S1 + S2**	**S3**	0.86	0.86	0.81	0.83	0.83	0.78
**S2**	**S3**	**S2 + S3**	**S1**	0.73	0.83	0.64	0.62	0.62	0.59
**S1**	**S3**	**S1 + S3**	**S2**	0.84	0.79	0.88	0.66	0.66	0.77
**Average**				**0.81**	**0.83**	**0.78**	**0.70**	**0.70**	**0.71**

Note: S1 means subject 1, S2 means subject 2, S3 means subject 3.

**Table 5 sensors-25-01876-t005:** Macro-F1 scores for multi-source M-FUDA and baseline models across cross-user tasks. Green background indicates the best accuracy for an individual task; blue background indicates best average accuracy; yellow background and bold font highlight the average performances.

Macro-F1									
Source Domains			Target Domain	Proposed Multi-Source Models		Combined-Source Models			
Source 1	Source 2	Source 3	Target 1	M-FUDA (MMD)	M-FUDA (MK-MMD)	M-FUDA (MMD)	CORAL [72]	Wasserstein [89]	MCD [59,88]
**S1**	**S2**	**S1 + S2**	**S3**	0.83	0.86	0.78	0.81	0.82	0.76
**S2**	**S3**	**S2 + S3**	**S1**	0.74	0.82	0.66	0.63	0.59	0.59
**S1**	**S3**	**S1 + S3**	**S2**	0.85	0.80	0.88	0.69	0.69	0.74
**Average**				**0.81**	**0.83**	**0.77**	**0.71**	**0.70**	**0.70**

Note: S1 means Subject 1, S2 means Subject 2, S3 means Subject 3.

**Table 6 sensors-25-01876-t006:** Micro-F1 scores for multi-source M-FUDA and baseline models across cross-user and cross-environment tasks. Green background indicates the best accuracy for an individual task; blue background indicates best average accuracy; yellow background and bold font highlight the average performances.

Micro-F1									
Source Domains			Target Domain	Proposed Multi-Source Models		Combined-Source Models			
Source 1	Source 2	Source 3	Target 1	M-FUDA (MMD)	M-FUDA (MK-MMD)	M-FUDA (MMD)	CORAL [72]	Wasserstein [89]	MCD [59,88]
**E1(S1)**	**E1(S2)**	**E1(S3)**	**E2(S12)**	0.69	0.65	0.67	0.62	0.62	0.67
**E1(S1)**	**E1 (S2)**	**E1(S3)**	**E3(S21)**	0.72	0.74	0.72	0.70	0.69	0.75
**E2(S11)**	**E2(S12)**	**E2(S13)**	**E1(S3)**	0.71	0.69	0.57	0.53	0.50	0.59
**E2(S11)**	**E2(S12)**	**E2(S13)**	**E3(S21)**	0.81	0.81	0.73	0.68	0.63	0.71
**E2(S11)**	**E2(S12)**	**E2(S13)**	**E3(S23)**	0.81	0.77	0.73	0.7	0.68	0.74
**E3(S21)**	**E3(S22)**	**E3(S23)**	**E1(S1)**	0.74	0.68	0.69	0.64	0.63	0.80
**E3(S21)**	**E3(S22)**	**E3(S23)**	**E1(S2)**	0.90	0.88	0.84	0.75	0.72	0.85
**E3(S21)**	**E3(S22)**	**E3(S23)**	**E1(S3)**	0.72	0.74	0.71	0.68	0.65	0.73
**E3(S21)**	**E3(S22)**	**E3(S23)**	**E2(S11)**	0.70	0.71	0.67	0.72	0.71	0.74
**E3(S21)**	**E3(S22)**	**E3(S23)**	**E2(S12)**	0.80	0.79	0.77	0.66	0.60	0.75
**E3(S21)**	**E3(S22)**	**E3(S23)**	**E2(S13)**	0.73	0.70	0.71	0.65	0.62	0.70
**Average**				**0.76**	**0.74**	**0.71**	**0.67**	**0.64**	**0.73**

Note: S1 means Subject 1, S2 means Subject 2, S3 means Subject 3, S11 means Subject 11, S12 means Subject 12, S13 means Subject 13, S21 means Subject 21, S22 means Subject 22, S23 means Subject 23, E1 means Environment 1, E2 means Environment 2, and E3 means Environment 3.

**Table 7 sensors-25-01876-t007:** Macro-F1 scores for multi-source M-FUDA and baseline models across cross-user and cross-environment tasks. Green background indicates the best accuracy for an individual task; blue background indicates best average accuracy; yellow background and bold font highlight the average performances.

Macro-F1									
Source Domains			Target Domain	Proposed Multi-Source Models		Combined-Source Models			
Source 1	Source 2	Source 3	Target 1	M-FUDA (MMD)	M-FUDA (MK-MMD)	M-FUDA (MMD)	CORAL [72]	Wasserstein [89]	MCD [59,88]
**E1(S1)**	**E1(S2)**	**E1(S3)**	**E2(S12)**	0.70	0.67	0.67	0.63	0.61	0.68
**E1(S1)**	**E1 (S2)**	**E1(S3)**	**E3(S21)**	0.68	0.71	0.72	0.69	0.68	0.74
**E2(S11)**	**E2(S12)**	**E2(S13)**	**E1(S3)**	0.68	0.65	0.54	0.52	0.50	0.55
**E2(S11)**	**E2(S12)**	**E2(S13)**	**E3(S21)**	0.81	0.81	0.73	0.69	0.62	0.70
**E2(S11)**	**E2(S12)**	**E2(S13)**	**E3(S23)**	0.81	0.78	0.74	0.7	0.67	0.73
**E3(S21)**	**E3(S22)**	**E3(S23)**	**E1(S1)**	0.72	0.66	0.66	0.62	0.61	0.78
**E3(S21)**	**E3(S22)**	**E3(S23)**	**E1(S2)**	0.90	0.86	0.83	0.75	0.73	0.85
**E3(S21)**	**E3(S22)**	**E3(S23)**	**E1(S3)**	0.69	0.72	0.68	0.67	0.64	0.71
**E3(S21)**	**E3(S22)**	**E3(S23)**	**E2(S11)**	0.71	0.72	0.68	0.72	0.70	0.75
**E3(S21)**	**E3(S22)**	**E3(S23)**	**E2(S12)**	0.81	0.79	0.78	0.65	0.59	0.76
**E3(S21)**	**E3(S22)**	**E3(S23)**	**E2(S13)**	0.75	0.72	0.72	0.64	0.62	0.71
**Average**				**0.75**	**0.74**	**0.70**	**0.66**	**0.63**	**0.72**

Note: S1 means Subject 1, S2 means Subject 2, S3 means Subject 3, S11 means Subject 11, S12 means Subject 12, S13 means Subject 13, S21 means Subject 21, S22 means Subject 22, S23 means Subject 23, E1 means Environment 1, E2 means Environment 2, and E3 means Environment 3.

**Table 8 sensors-25-01876-t008:** Micro-F1 scores for multi-source M-FUDA and baseline models across cross-atmospheric tasks. Green background indicates the best accuracy for an individual task; blue background indicates best average accuracy; yellow background and bold font highlight the average performances.

Micro-F1									
Source Domains			Target Domain	Proposed Multi-Source Models		Combined-Source Models			
Source 1	Source 2	Source 3	Target 1	M-FUDA (MMD)	M-FUDA (MK-MMD)	M-FUDA (MMD)	CORAL [72]	Wasserstein [89]	MCD [59,88]
**D6(S1)**	**D7(S1)**	**D8(S2)**	**D8(S1)**	0.70	0.68	0.69	0.57	0.58	0.68
**D7(S1)**	**D8(S1)**	**D6(S2)**	**D6(S1)**	0.78	0.75	0.80	0.71	0.68	0.89
**D6(S1)**	**D8(S1)**	**D7(S2)**	**D7(S1)**	0.76	0.73	0.75	0.68	0.66	0.70
**D6(S2)**	**D7(S2)**	**D8(S1)**	**D8(S2)**	0.67	0.66	0.63	0.58	0.60	0.62
**D7(S2)**	**D8(S2)**	**D6(S1)**	**D6(S2)**	0.74	0.67	0.74	0.67	0.65	0.73
**D6(S2)**	**D8(S2)**	**D7(S1)**	**D7(S2)**	0.75	0.72	0.73	0.69	0.67	0.72
**Average**				**0.73**	**0.70**	**0.72**	**0.65**	**0.64**	**0.72**

Note: S1 means Subject 1, S2 means Subject 2, D6 means day6, D7 means day7, and D8 means day8.

**Table 9 sensors-25-01876-t009:** Macro-F1 scores for multi-source M-FUDA and baseline models across cross-atmospheric tasks. Green background indicates the best accuracy for an individual task; blue background indicates best average accuracy; yellow background and bold font highlight the average performances.

Macro-F1									
Source Domains			Target Domain	Proposed Multi-Source Models		Combined-Source Models			
Source 1	Source 2	Source 3	Target 1	M-FUDA (MMD)	M-FUDA (MK-MMD)	M-FUDA (MMD)	CORAL [72]	Wasserstein [89]	MCD [59,88]
**D6(S1)**	**D7(S1)**	**D8(S2)**	**D8(S1)**	0.70	0.67	0.68	0.56	0.57	0.68
**D7(S1)**	**D8(S1)**	**D6(S2)**	**D6(S1)**	0.78	0.75	0.80	0.69	0.67	0.89
**D6(S1)**	**D8(S1)**	**D7(S2)**	**D7(S1)**	0.76	0.72	0.75	0.68	0.66	0.70
**D6(S2)**	**D7(S2)**	**D8(S1)**	**D8(S2)**	0.67	0.64	0.62	0.58	0.60	0.62
**D7(S2)**	**D8(S2)**	**D6(S1)**	**D6(S2)**	0.74	0.67	0.74	0.68	0.65	0.72
**D6(S2)**	**D8(S2)**	**D7(S1)**	**D7(S2)**	0.74	0.71	0.73	0.67	0.66	0.72
**Average**				**0.73**	**0.70**	**0.72**	**0.64**	**0.64**	**0.72**

Note: S1 means Subject 1, S2 means Subject 2, D6 means day6, D7 means day7, and D8 means day8.

**Table 10 sensors-25-01876-t010:** Average training time for multi-source M-FUDA and baseline models across cross-domain tasks. Green background indicates the best accuracy for an individual task; blue background indicates best average accuracy; yellow background and bold font highlight the average performances.

Training Time (seconds)						
Cross-Domain Tasks	Proposed Multi-Source Models		Combined-Source Models			
	M-FUDA (MMD)	M-FUDA (MK-MMD)	M-FUDA (MMD)	CORAL [72]	Wasserstein [89]	MCD [59,88]
**Cross-User**	521.97	632.49	370.29	324.26	312.34	353.91
**Cross-User + Cross-Environment**	504.23	615.13	430.47	409.12	395.56	426.89
**Cross-Atmospheric**	441.38	532.65	367.98	347.36	325.14	320.91
**Average**	**489.19**	**593.42**	**389.58**	**360.25**	**344.35**	**367.24**

Note: All the experimental values for training time are in seconds.

## Data Availability

The data presented in this study are available at the following uniform resource locator (URL) links: https://github.com/parisafm/CSI-HAR-Dataset, https://data.mendeley.com/datasets/v38wjmz6f6 and https://data.4tu.nl/datasets/575f95f7-abce-4be0-b4d6-29b4a683cf4c/1 (accessed on 13 January 2025).

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
