# Peer review of "Multi-Feature Unsupervised Domain Adaptation (M-FUDA) Applied to Cross Unaligned Domain-Specific Distributions in Device-Free Human Activity Classification"

_sensors, 2025, doi:10.3390/s25061876_

Round 1
Reviewer 1 Report
Comments and Suggestions for Authors
This paper presents a multi-source unsupervised domain adaptation (M-FUDA) technique for Human Activity Recognition (HAR), addressing cross-domain challenges such as variations in users, environments, and atmospheric conditions, while leveraging Wi-Fi Channel State Information (CSI) for device-free sensing and preserving privacy. Some issues should be addressed as follows.
1. While the paper proposes a novel approach, the innovation points are not clearly highlighted or discussed in sufficient detail. The contributions of the proposed method, particularly how M-FUDA advances the state of the art, should be more prominently emphasized.
2. The references cited in the paper are not sufficiently recent. Several important developments in the field, particularly in domain adaptation and device-free human activity recognition, are not mentioned.
3. Please provide a detailed explanation of how CSI is specifically applied to human activity recognition or device-free sensing in the context of this paper.
4. Are the data distributions from different source domains modeled using Wasserstein Distance? If so, please explain how this method contributes to improving the alignment between the source and target domains.
5. Terms like "domain-specific classifiers," "domain discrepancy minimization," and "multi-source environment" are used extensively but could be confusing to readers who are not experts in the field.
6. The figures are referenced but could benefit from more detailed captions. For instance, Figure 3 and Figure 4 are critical to understanding the architecture, but the descriptions might not fully explain how each part fits together.
7. The inconsistency in referring to "domain-specific feature extractors" versus "domain specific feature extractors" could lead to confusion and ambiguity for readers. Please ensure consistency in the usage of hyphenated terms throughout the paper.
Comments on the Quality of English LanguageThere are some writing issues in the paper, particularly with long sentences that could affect clarity. For instance, in Section 1, the sentence "The proposed model produces encouraging results surpassing the traditional domain..." is too long and should be broken down for better readability. In Section 8, the phrase "methods arranged in all sources combined together into a traditional single-source vs. target setting" is a long and redundant expression that can be revised for conciseness and clarity. The authors are encouraged to carefully review the entire paper to ensure that long and complex sentences are simplified for better flow and understanding.
Reviewer 2 Report
Comments and Suggestions for Authors
This paper gives a new method for adapting HAR from multi-source unsupervised domain adaptation algorithms using wifi channel CSI. And give many research results to evaluate the proposed method.
- In line 346, missing the reference ID .
- Reference 50 missing the page range.
- fig.5, whats the meaning of each colour labels? Why (a) has 7 labels from 0 to 6, (b) has 6 labels from 0 to 5 ,but (c) has 4 labels? And the fig.5 also did not give enough explanation for the results shown in the three figures. Table 1 seems list these labels, please give more explanations to show the links between the fig.5 and table 1.
- For table 4 and table 5, the improvement seems minor, for example, in table 4, under the proposed MMD, the result is 0.86, and under the source combined model, the MMD is 0.81, so this improvement seems not as expected. Besides, most of the micro-f1 scores are the same as the macro-f1 scores, why?
- How to evaluate the computing complexity?, this paper give multi-networks to train multi-source , the computing time many be long, such as in the table 10, and the computing complexity may be not low. So please give some explanations about the computing complexity .
- What’s the “Average” mean in last line in each table? Such as in table 1, the average means the average data number of the s1,s2 and s3? or the average of features ? or the average scores? So please give clear explanation about the AVEREAGE.
Reviewer 3 Report
Comments and Suggestions for Authors
Please revise the introduction more. There are more details can be provided including a bit of background on the capability of the sensing modality. Please include system workflow from dataset to classification phase.
Can authors highlight on the potential performance of their models in case that number of users is sufficiently large?
Can authors elaborate more on how they approached the uncertainties with classifying target classes near decision boundaries?
Please add comparison table, if applicable, to highlight the strengths of the proposed study as compared to previous studies.
In line 346, reference is missing.
The conclusion is too large and in one paragraph. Please revise.
